# Chemical Constituent Profiling of *Paecilomyces cicadae* Liquid Fermentation for *Astragli Radix*

**DOI:** 10.3390/molecules24162948

**Published:** 2019-08-14

**Authors:** Yuqi Wang, Xiaodan Mei, Zihan Liu, Jie Li, Xiaoxin Zhang, Shaoping Wang, Zikai Geng, Long Dai, Jiayu Zhang

**Affiliations:** 1School of Chinese Pharmacy, Beijing University of Chinese Medicine, Beijing 100029, China; 2School of Pharmacy, BIN ZHOU Medical University, Yantai 260040, China

**Keywords:** *Astragli Radix*, *Paecilomyces cicadae*, liquid fermentation, chemical constituents, UHPLC-LTQ-Orbitrap MS

## Abstract

*Astragli Radix* (AR) is one of the most popular traditional Chinese medicines with chemical constituents including flavonoids and saponins. As recently evidenced, some fungi or their fermentation liquid may have the potential to affect the bioactive constituents and different pharmacological effects of AR. Thus, the composition of fermented AR (FAR) produced by *Paecilomyces cicadae* (Miquel) Samson in liquid-state fermentation was investigated using a UHPLC-LTQ-Orbitrap mass spectrometer in both positive and negative ion modes. Firstly, the MS^n^ data sets were obtained based on a data-dependent acquisition method and a full scan–parent ions list–dynamic exclusion (FS-PIL-DE) strategy. Then, diagnostic product ions (DPIs) and neutral loss fragments (NLFs) were proposed for better constituent detection and structural characterization. Consequently, 107 constituents in total, particularly microconstituents in FAR and AR, were characterized and compared in parallel on the same LTQ–Orbitrap instrument. Our results indicated that AR fermentation with *Paecilomyces* significantly influenced the production of saponins and flavonoids, especially increasing the content of astragaloside IV. In conclusion, this research was not only the first to show changes in the chemical components of unfermented AR and FAR, but it also provides a foundation for further studies on the chemical interaction between microbiota and AR.

## 1. Introduction

Over the past several decades, accompanied by growing demand for traditional Chinese medicines (TCMs) and a gradual reduction of wild resources, improving the content of active ingredients and cultivating new varieties with high quality have become the most urgent tasks in the development of herbal resources. Recently, the application of TCMs by submerged fermentation of edible and pharmaceutical fungi has become a hot issue which opens up broad prospects for TCMs. Previous studies have shown that medicinal fungi can secrete important secondary metabolic products which degrade macromolecular material into small molecules [1,2]. By means of fermentation, TCMs can improve intrinsic conversion efficiency and new compound growth rates for increased therapeutic effect. Besides this, fermentation can also reduce the toxicity of TCMs containing typical compounds such as alkaloids, lactones, toxic glycosides, toxic proteins, anthraquinones, tannins, and heavy metals [3,4].

*Paecilomyces cicadae* (Miquel) Samson, as an entomogenous and medicinal fungus, is thought to be the anamorph stage of *Cordyceps cicadae* Shing. It is widely used as a tonic for nourishment as well as a functional food, and it has attracted considerable attention due to its wide range of nutritional and pharmacological activities, including immunomodulatory [5], antioxidation, anti-aging, anti-tumor [6], and anti-inflammation activity and ameliorating renal function [7].

Astragli Radix (AR), known as Huangqi in Chinese, is one of the most widely used traditional herbal medicines. It is the dried root of *Astragalus membranaceus* (Fisch.) Bge. var. *mongholicus* (Bge.) Hsiao or *Astragalus membranaceus* (Fisch.) Bge. AR contains relatively high quantities of saponins, flavonoids, polysaccharides, and some trace elements, which are known for their antibacterial, anti-inflammatory, analgesic, anti-cancer, anti-oxidant, and other pharmacological effects [8,9,10,11]. However, different processing methods (such as fermentation) may change the properties of this material.

Many studies have reported that natural macromolecular compounds exist in herbal medicines, including polysaccharides, flavonoid glycosides, and saponins, which promote pharmacological anti-tumor, anti-oxidant, and anti-inflammatory effects. However, most herbal macromolecular compounds cannot be digested and used by the human being in the absence of microbial fermentation [12]. For example, polysaccharides fermented by microbiota can be converted into short-chain fatty acids, which are easily digested and absorbed by the human body [13]. In addition, when red ginseng is fermented by *Bifidobacterium* H-1, Rg3 is transformed to Rh2, which has exhibited potent cytotoxicity against tumor cells [14]. Therefore, the aim of our present study was to develop liquid-state fermentation for AR by *Paecilomyces cicadae* (Miquel) Samson and to investigate whether this method leads to changes in the components contained in AR.

In order to obtain comprehensive knowledge of the compounds in the fermented AR (FAR), we further characterized its chemical constituents by way of ultra-high-performance liquid chromatography coupled with high-resolution mass spectrometry (UHPLC-HRMS). Moreover, the application of a full scan–parent ions list–dynamic exclusion (FS-PIL-DE) strategy coupled with diagnostic product ions (DPIs) and neutral loss fragments (NLFs) is proposed for better constituent detection and structural characterization [15,16]. Finally, the constituents, particularly microconstituents in FAR and AR, were characterized and compared in parallel on the same LTQ-Orbitrap instrument. 

## 2. Results

### 2.1. Establishment of the Analytical Strategy

In this study, a comprehensive and effective strategy is proposed to systematically screen and identify compounds on a UHPLC-LTQ-Orbitrap MS instrument. The analytic strategy roughly consisted of three steps. The first step was online data acquisition. A full mass scan was performed with a resolution of 30,000. Meanwhile, high-resolution extracted ion chromatography (HREIC) was adopted to extract the candidates from the high-quality, accurate raw mass data both in negative and positive ion modes. Secondly, PIL-DE and data-dependent acquisition methods were employed to obtain specific ESI-MS/MS datasets based on those screened candidates. Then, DPI and NLF techniques were used as supplementary tools for the selective detection of constituents that possess similar mass fragmentation behaviors to those of reference standards. Finally, the structures of the compounds were elucidated according to the accurate mass measurement, fragmentation patterns, diagnostic product ions, and literature reports. The general procedures of our strategy and approach are summarized in the diagram shown in Figure 1.

### 2.2. Establishment of the Data Acquisition Methods

By employing the full scan method, abundant data were generated with large amounts of invalid data. Thus, to reduce potential disturbances by irrelevant substances and avoid missing target drug compounds (especially trace compounds), HREIC was developed for compound detection [17]. The application of HREIC could intelligently filter the background and matrix-related ions from drug-related ions according to the accurate mass of [M-H]^−^ or [M+H]^+^ ions. The molecular weights and elemental compositions of compounds derived from the accurate mass measurements can also be readily predicted. As a result, the lower level of target compounds can be captured clearly. For a complicated system, FS was not an appropriate approach to obtain the entire MS/MS dataset due to the numerous potential candidates. Therefore, the PIL-DE method served as a supplementary method to obtain the MS/MS fragmentation of the microconstituents [15]. By means of the PIL-DE method, MS/MS acquisition of predictable constituents that have the same molecular weights could be triggered due to its superior sensitivity and selectivity.

### 2.3. Fragmentation Pattern Analysis and DPI Determination

To facilitate the structural elucidation of constituents in AR and FAR, sixteen standards, including eight astragalus saponins and eight flavonoids, were subsequently analyzed by UHPLC-LTQ-Orbitrap MS. All the standards exhibited [M-H]^−^ or [M+H]^+^ ions of sufficient intensity that could be isolated automatically and subjected to collision induced dissociation (CID)-MS/MS analysis. Mass Frontier v7.0 software (Thermo Scientific, Waltham, MA, USA) and manual elucidation were used to acquire comprehensive structural identification of these reference compounds.

In CID mode, compounds are often divided into two parts, such as product ions (emerge in ESI-MS/MS spectra due to their property of being easily ionized) and neutral fragments (observed in ESI-MS/MS spectra due to their mass difference and neutral characteristics) [18,19], which are complementary in structure. It is well documented that compounds with similar backbone structure exhibit comparable fragmentation patterns, resulting in certain diagnostic product ions (DPIs) and regular neutral loss fragments (NLFs). Consequently, the combination of DPIs and NLFs was helpful to rapidly performing the structural elucidation [20,21].

Eight astragalus saponin standards were subsequently analyzed firstly in the CID-MS/MS experiment. For instance, astragaloside I, isoastragaloside I, astragaloside II, isoastragaloside II, and astragaloside IV possess the same backbone structure, and their differences are limited to the quantity and position of acetyl groups connected to xylose. For instance, there are two acetyl groups on the 2 and 3 positions of xylose in astragaloside I, one acetyl group on the 2 position of xylose in astragaloside II, and zero acetyl groups in astragaloside IV. By comparing the MS/MS spectra of their product ions, some characteristic dissociation pathways of astragalus saponins could be summarized, which provided a basis for further characterization of the other candidates. Taking the negative ion mode as an example, all of the deprotonated ions could lose one glucosyl (C_6_H_10_O_5_∙) or xylose (C_5_H_8_O_4_∙) or even both of them in their ESI-MS spectra. Then, the base peak ions of [M-H-162]^−^, [M-H-132]^−^, and [M-H-294]^−^ could be formed. Owing to the special structure of the acetyl group (Ac), other characteristic fragment ions were also generated by the loss of 42 (Ac), 60 (Ac+H_2_O), and 84 (2Ac). These diagnostic product ions could be employed to ascertain the structural skeletons of astragalus saponins and simplify the following structural elucidation. 

In addition, we also selected eight flavonoids as subjects to determine their DPIs. Owing to the special structures of flavonoid glycosides, the base peak ion of [M-H-162]^−^ was usually produced via the loss of the glucose moiety in their ESI-MS^2^ spectra. Meanwhile, the other characteristic ions at [M-H-15]^−^, [M-H-18]^−^, [M-H-28]^−^, [M-H-29]^−^, [M-H-31]^−^, [M-H-43]^−^, [M-H-44]^−^, and [M-H-61]^−^ were yielded by losing CH_3_, H_2_O, CO, HCO, OCH_3_, CH_3_+CO, HCO + CH_3_, and H_2_O+CO+CH_3_ in negative mode. Therefore, the DPIs mentioned above could be utilized for deducing the structures of related compounds from abundant complex constituents.

### 2.4. Structural Assignment of Chemical Constituents in AR and FAR

Saponins and flavonoids are the major chemical constituents in AR. As a result, 107 compounds in total were detected and characterized from AR and FAR by way of UHPLC-LTQ-Orbitrap MS with the established strategy. Among these compounds, 42 were attributed to saponins while the remaining 65 were identified as flavonoids. The correlative data are summarized in Table 1 and Table 2, and the HREIC spectra of detected constituents are illustrated in Figure 2. The fragmentation patterns of representative saponins and flavonoids are shown in Appendix A.

#### 2.4.1. Structural Assignment of Saponins in AR and FAR

Most of the saponins in AR possess the same aglycone of cycloastragenol with different substituent groups, such as xylose, glucose, acetyl groups, and so on. They can be divided into type cyclolanostane cycloastragenol (**1–11**) or cyclolanostane cyclocanthogenin (**12–18**). Only a minority of saponins belonged to oleanane-type triterpenoids (**19**), the aglycones of which are attributed to soybean saponin B. There were 29 and 42 saponins screened and identified in FAR and AR, respectively, and their molecular formulae and chemical structures are shown in Table 3.

With retention times of 11.79 and 14.27 min, **A6** and **A16** afforded [M-H]^−^ ions at *m/z* 783.45612 and 783.45813 (C_41_H_67_O_14_, mass error within 5 ppm) in negative ion mode. Both of them produced the base peak ions at *m*/*z* 489 by neutral loss of the glucose and xylose moiety. Then, the product ion at *m*/*z* 489 further generated the predominant ion at *m*/*z* 453 by loss of 2H_2_O. Meanwhile, several important fragment ions at *m*/*z* 651 and *m*/*z* 621 were also observed due to the respective losses of xylose and glucose. Combined with standard substances, **A6** was positively characterized as isoastragaloside IV, while **A16** was speculated to be astragaloside IV.

**A18** produced its [M-H]^−^ ion at *m*/*z* 783.45654 (C_41_H_67_O_14_) with a mass error of 3.11 ppm. In the ESI-MS^2^ spectrum, further mass fragmentation resulted in *m*/*z* 489 [M-H-Glu-Xyl]^−^, *m*/*z* 621 [M-H-Glu]^−^, and *m*/*z* 453 [M-H-Xyl-Glu-2H_2_O]^−^, consistent with the characteristic fragmentation pathways of astragalus saponins. By comparing with the reference standard, **A18** was unambiguously identified as astragaloside III.

**A2**, **A11**, **A14**, **A22**, **A25**, and **A27**, which possessed a theoretical [M-H]^−^ ion at *m*/*z* 825.46419 (C_43_H_69_O_15_, mass error within 5 ppm), were eluted at 9.32, 12.59, 14.00, 16.67, 16.91, and 18.12 min, in order. In their ESI-MS^2^ spectra, the [M-H]^−^ ion at *m*/*z* 825 generated product ions at *m*/*z* 783, *m*/*z* 765, and *m*/*z* 633 by losing acetyl, acetyl+H_2_O, and xylose moieties. Among them, **A22** was positively identified as astragaloside II, and **A27** was unambiguously characterized as isoastragaloside II based on comparison of the MS/MS spectra and retention times with reference standards. The accurate mass weight and major product ions of **A2, A11**, **A14**, and **A25** were coincident with those of **A22**, indicating that they could be astragaloside II isomers.

**A30**, **A33**, **A36**, **A37**, and **A39** generated an identical [M-H]^−^ ion at *m*/*z* 867.47476 (C_45_H_71_O_16_) with mass errors within 5 ppm. All of their deprotonated molecular ions generated a series of fragment ions at *m*/*z* 807, *m*/*z* 765, and *m*/*z* 747, corresponding to [M-H-Ac-H_2_O]^−^, [M-H-2Ac-H_2_O]^−^, and [M-H-2Ac-2H_2_O]^−^. With the supplements of standard substances, **A37** was unambiguously characterized as astragaloside I, while **A39** was positively identified as isoastragaloside I. Therefore, **A30**, **A33**, and **A36** were determined to be astragaloside I isomers.

The ESI-MS^n^ spectra of **A16**, **A18**, **A22**, and **A37** are shown in Figure 3.

#### 2.4.2. Structural Assignment of Flavonoids in AR

AR contains a large number of flavonoids and glycosides, which can be divided into flavones (**1–5**), isoflavans (**6–10**), and isoflavones (**11–21**). Among these, the isoflavones are the most important group. Their molecular formulae and chemical structures are summarized in Table 4. In this work, 43 flavonoids in FAR and 47 flavonoids in AR were detected and characterized.

**B9** and **B20** possessed [M-H]^−^ ions at *m*/*z* 445.11481 and *m*/*z* 445.11575 (C_22_H_21_O_10_, mass errors 4.23 ppm and 4.35 ppm) in negative ion mode. DPIs, including [M-H-Glu]^−^ at *m*/*z* 283 and [M-H-Glu-CH_3_]^−^ at *m*/*z* 268, were also generated in their ESI-MS/MS spectra. By comparison with reference standards, **B9** was positively determined to be calycosin-7-glucoside, while **B20** was speculated to be a calycosin-7-glucoside isomer.

Two isomers, **B16** and **B22**, which displayed [M-H]^−^ ions at *m*/*z* 431.09955 and *m*/*z* 431.09961 (C_21_H_19_O_10_, mass error 3.28 ppm and 3.42 ppm), were eluted at 6.72 and 7.25 min, respectively. They yielded ESI-MS^2^ product ions at *m*/*z* 269 [M-H-Glu]^−^ and *m*/*z* 268 [M-2H-Glu]^−^. **B16** was positively identified as genistin based on the comparison of the ESI-MS/MS spectra and retention time with reference standards. Meanwhile, **B22** was predicted to be a genistin isomer.

**B26** was eluted at 7.52 min with an [M-H]^−^ ion at *m*/*z* 623.16339 (C_28_H_31_O_16_, mass error 4.379 ppm). The [M-H]^−^ ion at *m*/*z* 623 generated characteristic fragment ions at *m*/*z* 461, *m*/*z* 443, and *m*/*z* 299. The former was generated from the neutral loss of glucose (162 Da) from the [M-H]^−^ ion. The ion at *m*/*z* 461 was further fragmented to yield fragment ions at *m*/*z* 443 and *m*/*z* 299 by neutral loss of H_2_O (18 Da) and glucose (162 Da). Hence, **B26** was tentatively characterized as complanaruside.

In positive ion mode, **B37** gave rise to [M+H]^+^ ions at *m*/*z* 431.13263 with a retention time of 8.26 min. Its formula was speculated as C_22_H_23_O_9_ with a mass error of −2.38 ppm. DPIs at *m*/*z* 269 [M+H-Glu]^+^ and *m*/*z* 413 [M+H-Glu-H_2_O]^+^ were observed. With the addition of standard substances, **B37** was tentatively identified as ononin.

In negative ion mode, **B26**, which displayed [M-H]^−^ ions at *m*/*z* 429.11868 (C_22_H_21_O_9_, mass error 1.565 ppm), was eluted at 13.38 min. In the ESI-MS^2^ spectrum, it yielded product ions at *m*/*z* 411 [M-H-H_2_O]^−^ and *m*/*z* 267 [M-H-Glc]^−^. According to the retention times of reference substances, **B26** was unambiguously identified as ononin. Besides this, **B63** was eluted at 12.20 min with [M-H]^−^ ions at *m*/*z* 315.05081 (C_16_H_11_O_7_, mass error 2.79 ppm). On account of the neutral losses of CH_2_ and CH_3_, DPIs at *m*/*z* 301 and *m*/*z* 300 were respectively generated in its ESI-MS^2^ spectrum, which suggested the presence of a methoxy group. From the abovementioned analysis, **B63** could be deduced as isorhamnetin.

**B31**, **B36**, **B43**, **B46**, **B51**, **B61**, and **B64** were all observed with the same [M-H]^−^ ions at *m*/*z* 267.06683 (C_16_H_11_O_4_) with mass errors within 5 ppm. They all produced DPIs at *m*/*z* 252 [M-H-CH_3_]^−^ and *m*/*z* 253 [M-H-CH_2_]^−^ in the ESI-MS^2^ spectra, corresponding to the characteristic fragmentation pathways of a methoxy group. According to the standard references, compound **B64** was unambiguously characterized as formononetin, while the others were tentatively predicted to be formononetin isomers.

The ESI-MS^n^ spectra of B9, B16, B63, and B64 are shown in Figure 4.

#### 2.4.3. Comparative Analysis of Constituents in AR and FAR

To date, more than 100 compounds have been isolated and identified from AR. Saponins and flavonoids are considered the two most important constituents of AR for displaying bioactivities in vivo or in vitro [22]. Astragaloside I, isoastragaloside I, astragaloside II, isoastragaloside II, and astragaloside IV account for more than 80% of the total saponin content. It is noteworthy that variation of the saponin content among samples of different origins and parts, even in related preparations, is remarkable [23].

In our work, we found variation of saponins and flavonoids in FAR both in quality and amount, which was different from the former AR extract. The number of saponins species in FAR decreased from 42 to 29, while quantities of certain saponins, such as isoastragaloside IV, increased with the process of fermentation. As above, the flavone aglycones reduced in number from 30 to 25. The species and quantity of flavone glycosides changed obviously, even though the number was 17 in AR as well as in FAR (shown in Figure 5).

Moreover, the relative contents of some representative components changed greatly after fermentation (shown in Figure 6). In the process of fermentation, the contents of flavonoid glycosides—for instance, genistin, calycosin-7-glucoside, and complanaruside—dropped obviously. At the same time, the concentrations of certain saponins such as astragaloside I, astragaloside II, and isoastragaloside I decreased after fermentation, too. This result suggests that fermentation can accelerate the conversion of saponin glycosides into saponin aglycons and the hydrolysis of flavonoid glycosides to monoglycosides or aglycones. Besides this, owing to the presence of methoxyl groups, flavonoids were extremely unstable under the fermentation process. Thus, vast amounts of compounds may be altered into isomers during the fermentation process.

Noteworthily, the contents of astragaloside IV and isoastragaloside IV were significantly increased after fermentation, which means that the production of astragaloside IV was significantly higher than its consumption. It is also worth mentioning that astragaloside IV, noted for the quality control evaluation of AR in the Chinese Pharmacopeia, exhibits protective effects on cardiovascular disease, focal cerebral ischemia/reperfusion, liver cirrhosis, pulmonary disease, and diabetic nephropathy [24]. Although its content is relatively low in crude drugs, other astragalosides tend to be transformed into astragaloside IV in the fermentation process, which indicates that FAR may contribute to getting the necessary amount for the desired therapeutic effect. The probable transformations of astragaloside IV are illustrated in Figure 7.

A few issues remain with this study. For example, fermentation induced a significant difference in compounds in FAR, but no specific transforming relationship was shown. The structures of newly generated constituents in AR by fermentation of *Paecilomyces cicadae* still remain obscure, but our findings encourage a much more in-depth analysis and structural elucidation.

## 3. Conclusions

In the present study, an effective strategy was established for the rapid screening and identification of target constituents in AR and FAR using FS-PIL-DE acquisition coupled to DPI analysis on a hybrid LTQ-Orbitrap MS in both positive and negative ion modes. A total of 107 compounds was preliminarily identified, including 42 saponins and 65 flavonoids. Our results indicated that AR fermentation with *Paecilomyces* significantly influenced the production of saponins and flavonoids. Among these compounds, the saponins were remarkedly reduced in connection with fermentation. This may be due to the degradation of saponins or flavonoid glycosides by hydrolytic enzymes, allowing the deglycosylated main backbone of glucoside to be divided into aglycone and oligosaccharides. This is the first study to show the changes in chemical components of unfermented AR and FAR, and it provides a foundation for further studies on the chemical interaction between microbiota and AR.

## 4. Materials and Methods

### 4.1. Materials and Reagents

*Astragalus membranaceus* (Fisch.) Bge. var. mongholicus (Bge.) Hsiao was obtained from Beijing Bencao Fangyuan Pharmaceutical Co., Ltd. (Beijing, China) and verified by Professor Yuan Zhang (Beijing University of Chinese Medicine, China). *Paecilomyces cicadae* (Miquel) Samson (No. cfcc81169) was provided by China Forestry Culture Collection Center (Beijing, China). Sixteen reference compounds, including astragaloside I, astragaloside II, astragaloside III, astragaloside IV, isoastragaloside I, isoastragaloside II, isoastragaloside IV, β-D-Glucopyranoside, (3β, 6α, 16β, 20R, 24S)-3-[(3,4-di-O-acetyl-β-D-xylopyranosyl)oxy]-20,24-epoxy-16,25-dihydroxy-9,19-cyclolanostan-6-yl, calycosin, calycosin-7-glucoside, formononetin, ononin, astraisoflavan-7-Oβ-D-glucoside, genistin, complanaruside, and isorhamnetin, were all purchased from Chengdu Must Biotechnology Co. Ltd. (Sichuan, China). Their structures were fully elucidated by comparing their spectra with the published literature. Their purities were acceptable (≥98%) according to the requirements for HPLC-UV or HPLC-ELSD analysis.

HPLC-grade acetonitrile and formic acid (FA) were purchased from Thermo Fisher Scientific (Fair Lawn, NJ, USA). All other chemicals of analytical grade were available at the work station, Beijing Chemical Works (Beijing, China). Deionized water used throughout the experiment was purified by a Milli-Q Gradient Å 10 System (Millipore, Billerica, MA, USA). Grace Pure^TM^ SPE C_18_-Low solid-phase extraction (SPE) cartridges (200 mg/3 mL, 59 m, 70 Å) were purchased from Grace Davison Discovery Science (Deerfield, IL, USA).

### 4.2. Fermentation of AR

AR was hot-air-dried for 2 days and then ground into a powder through a 100-mesh screen form using a blender. Laboratory-scale fermentation using AR was carried out in a 500 mL shake flask with a 250 mL working volume including 50 g of AR powder. A quantity of 50 g of AR powder was dissolved with 250 mL of distilled water and extracted at 121 °C for 15 min by autoclaving. *Paecilomyces cicadae* (Miquel) Samson grown at 5% (*v*/*v*) in PDA liquid medium was used as an inoculum. The mixture was fermented at 28 °C for 7 days on a rotatory shaker at 120 rpm∙min^−1^. Samples were taken on the 14th day of fermentation for analyses. Unfermented AR was ground into a powder using a 100-mesh screen, inoculated into distilled water without *Paecilomyces cicadae* (Miquel) Samson, and cultured for 7 days at 28 °C under aerobic conditions.

A volume of 1 mL of AR and FAR solution was added into an SPE cartridge pretreated with 5 mL methanol and 5 mL deionized water, in that order. Afterwards, the SPE cartridges were successively washed with 3 mL deionized water and 3 mL methanol, separately. The methanol eluate was evaporated to dryness by water bath at 70 °C. Then, the residue was re-dissolved in 200 µL methanol solution and centrifuged for 30 min (13,500 rpm, 4 °C). The supernatant was used for subsequent analysis.

### 4.3. UHPLC-LTQ-Orbitrap MS Analysis

#### 4.3.1. Instrument and Conditions

UHPLC analysis was performed on a DIONEX Ultimate 3000 UHPLC system (Thermo Fisher Scientific, Waltham, MA, USA), equipped with a binary pump, an auto-sampler, a column compartment, and an electrospray ionization source. The chromatographic separation was carried out at 40 °C using a Waters ACQUITY HSS T3 column (2.1 × 100 mm i.d., 1.8 μm; Waters Corporation, Milford, MA, USA). The mobile phase consisted of 0.1% FA aqueous solution (A) and acetonitrile (B) at a flow rate of 0.3 mL/min, and the linear gradient procedure was as follows: 0–6 min, 8%–30% B; 6–14 min, 30%–40% B; 14–20.5 min, 40%–50% B; 20.5–26 min, 30%–40% B; 26–30 min, 40%–95% B. The injection volume was 2 μL.

HRMS and MS/MS spectra were obtained using LTQ-Orbitrap MS with optimized operating parameters set as follows. Positive ion mode: sheath gas (nitrogen) flow rate of 40 arb, auxiliary gas (nitrogen) flow rate of 20 arb, capillary temperature of 350 °C, spray voltage of 4.0 kV, capillary voltage of 25 V, tube lens voltage of 110 V. Negative ion mode: sheath gas (nitrogen) flow rate of 40 arb, auxiliary gas (nitrogen) flow rate of 20 arb, capillary temperature of 350 °C, spray voltage of 3.0 kV, capillary voltage of −35 V, tube lens voltage of −110 V. The metabolites were detected by full-scan mass analysis from *m/z* 100 to *m/z* 1200 with a resolution of 30,000 in positive and negative ion modes. The collision energy for collision induced dissociation (CID) was adjusted to 40% of the maximum. Dynamic exclusion (DE) was used to prevent duplication. The repeat count was set to 5, and the dynamic repeat time was 30 s with a dynamic exclusion duration of 60 s. In addition, MS^n^ stages of the obtained datasets were employed using the PIL-DE dependent acquisition mode.

#### 4.3.2. Data Processing

A Thermo Xcalibur 2.1 (Thermo Scientific) workstation was used for data acquisition and data processing. In order to acquire as many fragment ions as possible, we selected the peaks with intensity over 10,000 for negative ion mode and over 40,000 for positive ion mode to identify components in AR and FAR. Based on the accurate mass, potential element compositions, and occurrence of possible reactions, the predicted atoms for chemical formulae of all the deprotonated and protonated molecular ions were set as follows: C [0–50], H [0–90], O [0–30], and ring double bond (RDB) equivalent value [0–15]. The maximum mass errors between the measured and calculated values were fixed within 5 ppm. All the relevant data, including peak number, retention time, accurate mass, the predicted chemical formula, and corresponding mass error, were recorded.

## Figures and Tables

**Figure 1 molecules-24-02948-f001:**
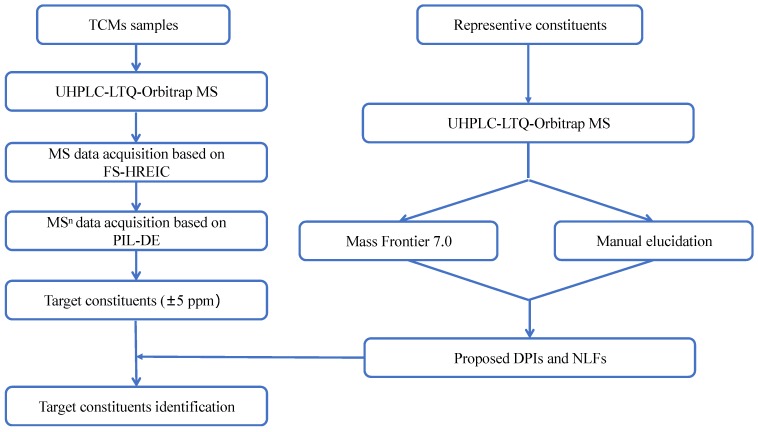
Summary diagram of the presently developed strategy and methodology.

**Figure 2 molecules-24-02948-f002:**
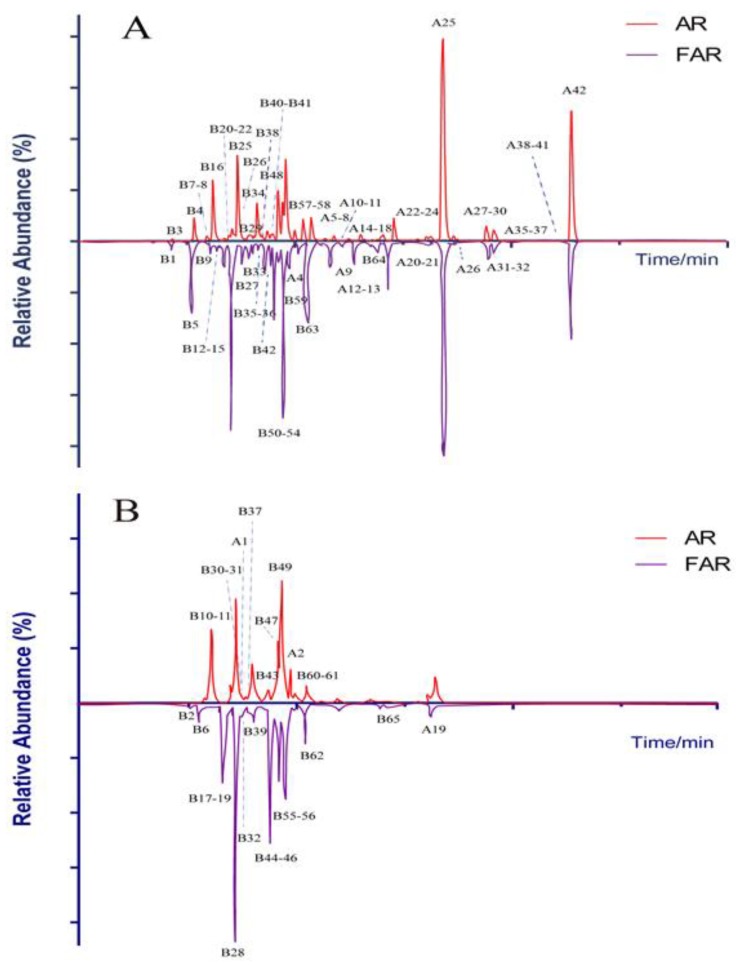
High-resolution extracted ion chromatograms for 107 compounds in AR and FAR. (**A**) Results of negative ion mode; (**B**) results of positive ion mode.

**Figure 3 molecules-24-02948-f003:**
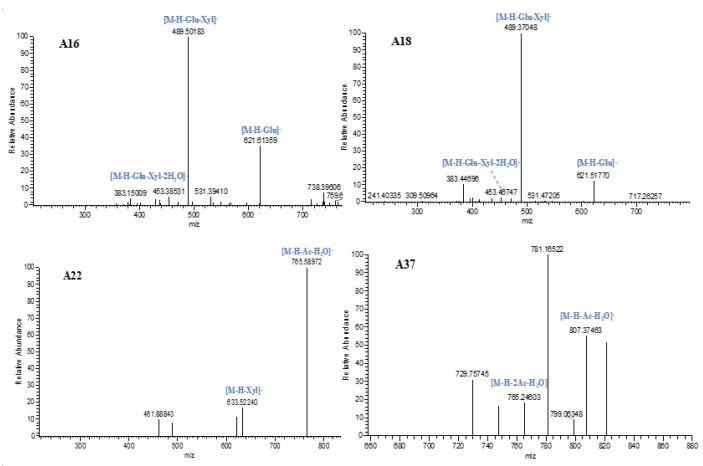
The ESI-MS^n^ spectra of **A16**, **A18**, **A22**, and **A37**.

**Figure 4 molecules-24-02948-f004:**
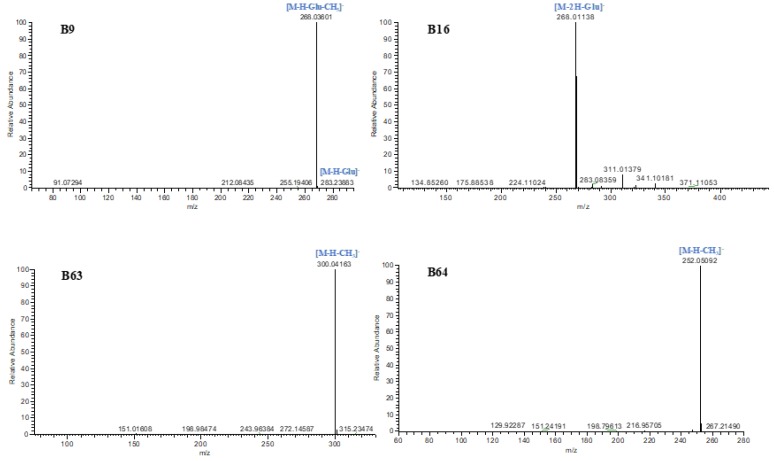
The ESI-MS^n^ spectra of **B9**, **B16**, **B63**, and **B64**.

**Figure 5 molecules-24-02948-f005:**
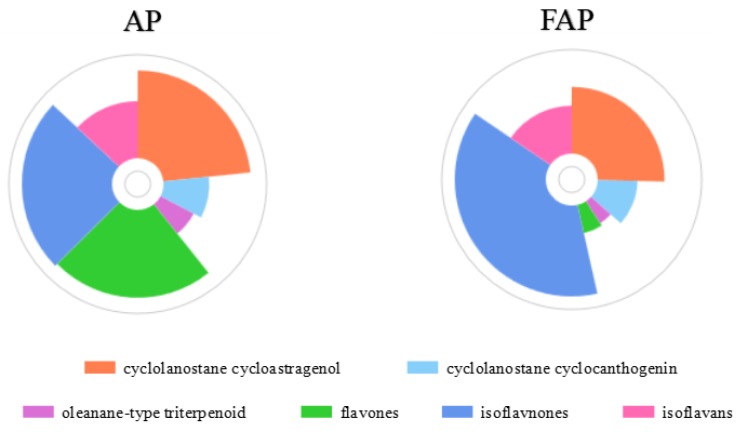
The classification of constituents in AR and FAR.

**Figure 6 molecules-24-02948-f006:**
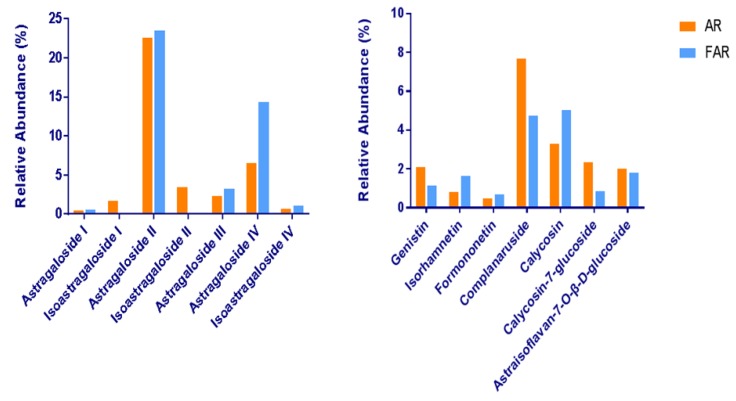
The changes in representative constituent contents in AR and FAR.

**Figure 7 molecules-24-02948-f007:**
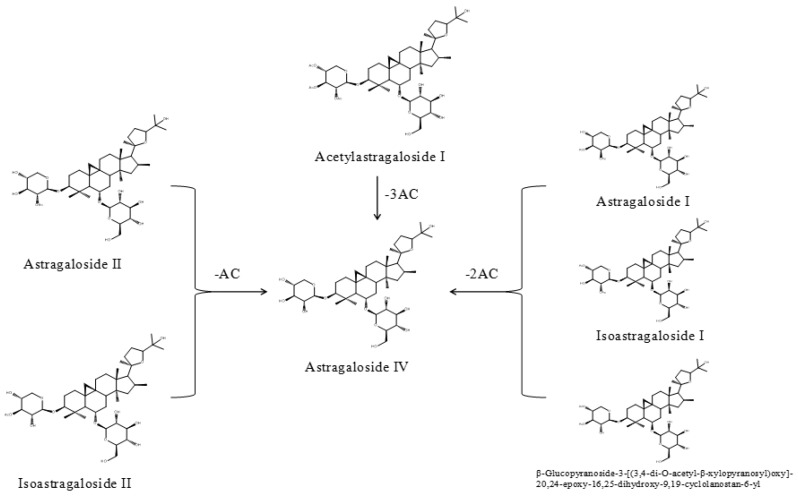
The probable transformations of astragaloside IV.

**Table 1 molecules-24-02948-t001:** Summary of identified saponins in Astragli Radix (AR) and fermented AR (FAR).

Peak	t_R_ /min	Ion Mode	Formula	Theoretical Mass *m/z*	Experimental Mass *m/z*	Error (ppm)	MS^2^/MS^3^ Fragment Ions	Identification	FAR	AR
**A1**	7.57	P	C_48_H_79_O_18_	943.52664	943.52582	−0.288	MS^2^[943]:925(100),927(76),486(30),859(13),927(13),845(10),827(6)	Soyasaponin I/isomer	-	+
**A2**	9.32	P	C_43_H_71_O_15_	827.47875	827.47443	−3.218	MS^2^[827]:709(100),809(10),691(9),768(4),737(2),695(2),577(2),335(2),467(1)	Acetylastragaloside II isomer	+	++
**A3**	10.43	P	C_38_H_63_O_11_	695.43704	695.43274	−3.391	MS^2^[695]:577(100),677(12),559(9),583(5),576(2)	Mongholicoside II	+	++
**A4**	10.76	N	C_47_H_77_O_19_	945.50700	945.50916	4.023	MS^2^[945]:783(100),489(3),621(2),765(1),651(1) MS^3^[783]:489(100),621(53),383(35),651(26)	Agroastragaloside V	+	++
		P	C_47_H_79_O_19_	947.52155	947.52026	−0.788	MS^2^[947]:437(100),455(52),419(42),473(22),587(21),569(16),599(12),605(11),617(7) MS^3^[437]:419(100),401(18)			
**A5**	11.61	N	C_41_H_69_O_14_	785.46983	785.47198	4.834	MS^2^[785]:491(100),623(26),489(15),435(6),649(6),717(4),741(1)	Cyclocanthoside E/isomer	+	++
**A6***	11.79	N	C_41_H_67_O_14_	783.45363	783.45612	4.578	MS^2^[783]:489(100),621(46),651(36),383(15),453(11),515(8),471(6)	Isoastragaloside IV	++	+
**A7**	11.82	N	C_42_H_69_O_15_	813.46474	813.46747	3.375	MS^2^[813]:745(100),767(67),771(63),607(37),651(20),387(25)	Astramembranoside A/isomer	+	++
**A8**	11.89	N	C_41_H_69_O_14_	785.46983	785.46277	−4.892	MS^2^[785]:491(100),623(58),489(44),653(30),717(21),737(20)	Cyclocanthoside E/isomer	+	++
**A9**	12.42	N	C_42_H_69_O_15_	813.46474	813.46729	3.154	MS^2^[813]:651(100),687(64),745(47),767(44),473(26),707(23)	Astramembranoside A/isomer	+	++
		P	C_42_H_71_O_15_	815.47930	815.47729	−1.788	MS^2^[815]:437(100),455(48),419(41),473(29),599(20),653(7),278(6),437(6),467(6),745(5) MS^3^[437]:419(100),351(26),175(22),215(16),253(16),167(10)			
**A10**	12.55	N	C_41_H_69_O_14_	785.46983	785.47180	4.605	MS^2^[785]:491(100),623(24),717(13),747(4),629(4),701(3)	Cyclocanthoside E/isomer	-	++
**A11**	12.59	N	C_43_H_69_O_15_	825.46419	825.46735	3.151	MS^2^[825]:765(100),783(45),757(17),787(12),779(11),673(5),401(4)	Astragaloside II isomer	+	++
**A12**	13.32	N	C_36_H_61_O_11_	669.42248	669.42383	4.468	MS^2^[669]:623(100),533(46),465(39),367(29),651(18)	Mongholicoside A /isomer	+	++
**A13**	13.79	N	C_43_H_71_O_15_	827.48039	827.48138	3.181	MS^2^[827]:759(100),767(39),784(36),357(34),781(33),616(24)785(22),770(20)	Agroastragaloside II	+	++
**A14**	14.00	N	C_43_H_69_O_15_	825.46419	825.46710	4.849	MS^2^[825]:633(100),763(38),765(34),645(27),486(17),643(15),783(15)	Astragaloside II isomer	-	+
**A15**	14.07	N	C_36_H_61_O_11_	669.42248	669.42377	4.378	MS^2^[669]:623(100),601(57),397(26),533(21),601(20),625(19),641(17),651(15)	Mongholicoside A /isomer	+	++
**A16***	14.27	N	C_41_H_67_O_14_	783.45363	783.45813	4.144	MS^2^[783]:489(100),621(14),383(11),453(4)	Astragaloside IV	++	+
**A17**	14.55	N	C_41_H_69_O_14_	785.46983	785.46313	−4.433	MS^2^[785]:490(100),489(79),491(28),623(16),383(13),621(11)	Cyclocanthoside E/isomer	+	++
**A18***	14.59	N	C_41_H_67_O_14_	783.45363	783.45654	3.115	MS^2^[783]:489(100),383(13),621(12),453(4),401(2),472(2),381(2)	Astragaloside III	++	+
**A19**	16.10	N	C_51_H_81_O_21_	1029.52758	1029.52173	−4.619	MS^2^[1029]:985(100),984(18),967(2)	Agroastragaloside III	+	++
		P	C_51_H_83_O_21_	1031.54214	1031.54199	−0.141	MS^2^[1031]:984(100),494(57),558(52),331(50),667(49),936(48),323(47),482(46),300(45)			
**A20**	16.29	N	C_48_H_77_O_18_	941.51209	941.50549	−5.259	MS^2^[941]:923(100),524(56),873(36),923(32),615(27),523(26),879(20),456(18),	Soyasaponin I/isomer	+	++
**A21**	16.31	N	C_47_H_73_O_17_	909.48532	909.48804	4.192	MS^2^[909]:891(100),523(50),569(49),613(49),455(35),701(31),435(18),757(16)	Acetylastragaloside I /isomer	+	++
**A22***	16.67	N	C_43_H_69_O_15_	825.46419	825.46796	3.890	MS^2^[825]:765(100),633(17),621(11),461(10),489(8)	Astragaloside II	++	+
		P	C_43_H_71_O_15_	827.47875	827.47729	−1.762	MS^2^[827]:269(100),592(67),629(66),351(64),296(63),633(60),709(60),247(59),277(57)			
**A23**	16.74	N	C_36_H_59_O_11_	667.40683	667.40820	4.512	MS^2^[667]:649(100),449(82),621(81),299(80),450(74),485(54)	Mongholicoside B	+	++
**A24**	16.85	N	C_48_H_77_O_18_	941.51209	941.51392	3.694	MS^2^[941]:923(100),525(73),615(51),744(49),879(41),457(40),795(37),437(35),597(16) MS^3^[923]:525(100),733(55),879(47),437(44),597(28),457(21)	Soyasaponin I/isomer	+	++
		P	C_48_H_79_O_18_	943.52664	943.52496	−1.200	MS^2^[943]:599(100),797(88),441(79),423(48),617(28),581(23),520(10),269(8),454(8),867(8),448(8)			
**A25**	16.91	N	C_43_H_69_O_15_	825.46419	825.46631	3.892	MS^2^[825]:783(100),607(32),765(30),735(16),795(10),697(9),758(7)	Astragaloside II isomer	-	+
**A26**	17.53	N	C_42_H_69_O_15_	813.46474	813.46686	4.625	MS^2^[813]:767(100),274(73),677(18)	Astramembranoside A/isomer	+	++
**A27***	18.12	N	C_43_H_69_O_15_	825.46419	825.46643	4.037	MS^2^[825]:765(100),633(19),717(24),495(20),351(5)	Isoastragaloside II	-	+
**A28**	18.80	N	C_42_H_65_O_14_	793.43853	793.44080	4.937	MS^2^[793]:631(100),725(24),657(8),724(7),747(5),718(5),697(4)	Huangqiyenin E/isomer	-	+
**A29**	18.83	N	C_42_H_69_O_15_	813.46474	813.46692	4.699	MS^2^[813]:767(100),677(11),795(1)	Astramembranoside A/isomer	+	++
**A30**	18.94	N	C_45_H_71_O_16_	867.47476	867.47766	4.608	MS^2^[867]:807(100),765(61),821(24),731(24),821(23)	Astragaloside I isomer	+	++
**A31**	19.16	N	C_47_H_73_O_17_	909.48532	909.48846	4.654	MS^2^[909]:891(100),455(86),569(33),523(29),613(28),407(28),763(26),773(22)	Acetylastragaloside I/isomer	+	++
**A32**	19.21	N	C_48_H_77_O_18_	941.51209	941.50427	−4.555	MS^2^[941]:923(100),523(44),879(37),614(36),523(36),613(32),732(31)	Soyasaponin I	+	++
**A33**	19.34	N	C_45_H_71_O_16_	867.47476	867.47766	4.608	MS^2^[867]:821(100),799(34),731(23),343(16),787(11)	Astragaloside I isomer	-	+
**A34**	19.44	P	C_48_H_79_O_18_	943.52664	943.52161	−4.750	MS^2^[943]:796(100),598(88),439(30),597(27),795(13)	Soyasaponin I/isomer	-	+
**A35**	20.25	N	C_42_H_65_O_14_	793.43853	793.44073	4.849	MS^2^[793]:455(100),613(86),435(85),391(45)	Huangqiyenin E/isomer	-	+
**A36**	20.34	N	C_45_H_71_O_16_	867.47476	867.47662	3.410	MS^2^[867]:807(100),799(52),765(51),731(44),825(43),731(29)	Astragaloside I isomer	-	+
**A37***	20.95	N	C_45_H_71_O_16_	867.47476	867.47943	4.649	MS^2^[867]:781(100),807(55),821(51),765(18),747(16)	Astragaloside I	++	+
**A38**	22.12	N	C_45_H_73_O_16_	869.49096	869.49335	4.644	MS^2^[869]:823(100),801(46),599(18),785(15),536(11),731(10),741(8),705(8)	Agroastragaloside I	-	+
**A39***	22.72	N	C_45_H_71_O_16_	867.47476	867.47711	3.974	MS^2^[867]:807(100),747(38),685(29),717(19),765(16),749(10)	Isoastragaloside I	-	+
**A40**	22.77	N	C_48_H_73_O_19_	953.47570	953.47968	3.898	MS^2^[953]:909(100),849(3),867(2),807(1) MS^3^[909]:849(100),807(87),765(80),867(56),747(42)	Malonylastragaloside I	+	++
**A41**	22.87	N	C_47_H_73_O_17_	909.48532	909.48846	4.654	MS^2^[909]:849(100),867(27),765(24),801(14),807(12),747(11),867(10)	Acetylastragaloside I/isomer	+	++
**A42***	23.79	N	C_45_H_71_O_16_	867.47476	867.47754	4.470	MS^2^[867]:807(100),287(75),765(73),645(65),799(63),723(34),850(31)	β-D-glucopyranoside,(3β,6α,16β,20R,24s)-3-[(3,4-di-O-acetyl-β-D-xylopyranosyl)oxy]-20,24-epoxy-16,25-dihydroxy-9,19-cyclolanostan-6-yl	-	+

**Table 2 molecules-24-02948-t002:** Summary of identified flavonoids in AR and FAR.

Peak	t_R_ /min	Ion mode	Formula	Theoretical Mass *m/z*	Experimental Mass *m/z*	Error (ppm)	MS^2^/MS^3^ fragment ions	Identification	FAR	AR
**B1**	4.37	N	C_29_H_37_O_16_	641.20926	641.21063	4.708	MS^2^[641]:479(100),317(75),595(35),611(30),623(26),379(24),610(22)	5′-hydroxy-isomucronulatol-2′,5′-di-o-glucoside	+	++
**B2**	4.47	P	C_24_H_25_O_12_	505.13460	505.13318	−1.727	MS^2^[505]:333(100),335(41),373(26),438(21),281(21),343(13),282(11),317(9),181(7),487(6)	Neocomplanoside/isomer	+	++
**B3**	4.76	N	C_28_H_31_O_16_	623.16231	623.16388	3.165	MS^2^[623]:299(100),284(31),604(7),283(6),456(6),605(5),443(5),255(4)	Complanatuside isomer	-	+
**B4**	5.23	N	C_28_H_31_O_16_	623.16231	623.16364	4.780	MS^2^[623]:299(100),284(32),443(10),240(4),461(3),577(2),605(2),211(2),239(2)	Complanatuside isomer	-	+
**B5**	5.35	N	C_22_H_21_O_11_	461.10948	461.11050	3.773	MS^2^[461]:299(100),284(9) MS^3^[299]:284(100)	Kaempferol-4′-methylether-3-D-glucoside	+	++
		P	C_22_H_23_O_11_	463.12404	463.12265	−1.809	MS^2^[463]:445(100),401(4),344(4),234(3),431(1),301(1)			
**B6**	5.53	P	C_16_H_17_O_5_	289.10760	289.10645	−2.076	MS^2^[289]:271(100),270(91),221(78),205(76),233(32),261(17)	(3R)-7,2′,3′-trihydroxy-4′-methoxy isoflavonone/isomer	+	-
**B7**	5.99	N	C_16_H_11_O_5_	283.06175	283.06198	3.642	MS^2^[283]:268(100),269(3) MS^3^[268]:240(100),239(49),334(46),211(44),195(23)	Calycosin isomer	-	+
		P	C_16_H_13_O_5_	285.07630	285.07529	−1.614	MS^2^[285]:270(100),253(44),225(18),137(7),271(5),257(3)			
**B8**	6.19	N	C_24_H_23_O_12_	503.12005	503.12112	4.401	MS^2^[503]:299(100),284(23),443(4),461(2),484(1),240(1)	Neocomplanoside/isomer	-	+
**B9** *	6.20	N	C_22_H_21_O_10_	445.11457	445.11481	4.239	MS^2^[445]:283(100),268(17)	Calycosin-7-glucoside	+	++
		P	C_22_H_23_O_10_	447.12912	447.12695	−3.630	MS^2^[447]:285(100),334(8),403(2),306(1),241(1) MS^3^[285]:270(100),253(41),225(17),137(7),229(5),211(5)			
**B10**	6.26	P	C_16_H_13_O_5_	285.07630	285.07678	3.613	MS^2^[285]:270(100),253(43),225(19),137(9) MS^3^[270]:137(100),253(57),214(31),242(13),134(12),213(12)	Calycosin isomer	-	+
**B11**	6.34	P	C_17_H_15_O_6_	315.08686	315.08603	−0.903	MS^2^[315]:300(100),283(20),255(8),167(5),259(4),301(2),287(2),175(2)	7,3′-dihydroxy-8,4-dimethoxyisoflavone/isomer	-	+
**B12**	6.22	N	C_22_H_21_O_12_	477.10440	477.10532	4.382	MS^2^[477]:315(100),301(18),300(14),347(13),313(11),458(5),278(4)	isorhamnetin-3-D- glucoside	+	++
**B13**	6.35	N	C_17_H_13_O_6_	313.07231	313.07236	4.415	MS^2^[313]:298(100),285(2),295(1),287(1),283(1)	7,3′-dihydroxy-8,4-dimethoxyisoflavone/isomer	+	++
**B14**	6.49	N	C_22_H_21_O_9_	429.11965	429.12024	4.200	MS^2^[429]:252(100),253(25),295(7),267(3),411(2),361(2),383(2),231(2)	Ononin isomer	+	-
**B15**	6.56	N	C_23_H_23_O_11_	475.12513	475.12579	4.845	MS^2^[475]:298(100),283(50),299(14),297(5),255(4),443(4),194(4),277(3)	Odoratin-7-O-*β*-D-glucoside/isomer	+	-
**B16** *	6.72	N	C_21_H_19_O_10_	431.09727	431.09955	3.281	MS^2^[431]:268(100),269(62),311(7),341(2),283(2)	Genistin	+	++
**B17**	6.82	N	C_23_H_23_O_11_	475.12513	475.12619	3.687	MS^2^[475]:299(100),284(13),298(7),460(5),283(2),297(2),431(1) MS^3^[299]:284(100),240(1)	Odoratin-7-O-*β*-D- glucoside/isomer	+	-
		P	C_23_H_25_O_11_	477.13969	477.13742	−3.601	MS^2^[477]:301(100),345(10),199(10),183(8),453(7) MS^3^[301]:286(100),269(33),153(29),245(15),241(14),152(6),223(5),175(2),273(2),123(1)			
**B18**	6.99	N	C_16_H_11_O_5_	283.06175	283.06192	4.430	MS^2^[283]:268(100),269(4) MS^3^[268]:240(100),239(63),211(55),224(40),184(28),195(27)	Calycosin isomer	++	+
		P	C_16_H_13_O_5_	285.07630	285.07529	−1.614	MS^2^[285]:270(100),253(43),225(20),285(17),137(9),229(7),286(4),257(3),181(2)			
**B19**	7.02	N	C_16_H_15_O_5_	287.09305	287.09225	2.961	MS^2^[287]:243(100),203(53),201(19),219(11),259(9),173(7),157(5)	(3R)-7,2′,3′-trihydroxy-4′-methoxy isoflavonone/isomer	+	-
		P	C_16_H_17_O_5_	289.10760	289.10651	−1.868	MS^2^[289]:271(100),184(8),252(8),166(7),205(4),182(2)			
**B20**	7.10	N	C_22_H_21_O_10_	445.11457	445.11575	4.351	MS^2^[445]:283(100),268(17) MS^3^[283]:268(100)	Calycosin-7-glucoside isomer	-	+
**B21**	7.16	N	C_17_H_13_O_6_	313.07231	313.07230	3.224	MS^2^[313]:298(100),181(17),245(8),137(6),295(6),269(5),139(5),131(3),194(3)	7,3′-dihydroxy-8,4-dimethoxyisoflavone/isomer	+	++
**B22**	7.25	N	C_21_H_19_O_10_	431.09892	431.09961	3.421	MS^2^[431]:268(100),269(48),311(8),413(6),341(4),323(2),412(2)	genistin isomer	-	+
**B23**	7.28	P	C_17_H_15_O_6_	315.08686	315.08575	−1.792	MS^2^[315]:300(100),283(19),255(9),269(8),297(5),167(5),259(4),138(3)	7,3′-dihydroxy-8,4-dimethoxyisoflavone/isomer	+	-
**B24**	7.36	N	C_15_H_9_O_5_	269.04610	269.04352	−3.456	MS^2^[269]:241(100),240(58),225(48),197(25),185(20),213(15)	5,7,4′-trihydroxy- isoflavonone/isomer	-	+
**B25**	7.38	N	C_16_H_11_O_5_	283.06175	283.06189	4.324	MS^2^[283]:268(100),269(1) MS^3^[268]:240(100),211(58),239(55),224(40),195(25)	Calycosin isomer	+	++
		P	C_16_H_13_O_5_	285.07630	285.07571	−0.140	MS^2^[285]:270(100),253(43),225(20),285(14),137(8),229(7),286(5),257(3),181(2),197(1) MS^3^[270]:137(100),253(54),214(33),213(14),134(13),242(12)			
**B26** *	7.52	N	C_28_H_31_O_16_	623.16231	623.16339	4.379	MS^2^[623]:461(100),299(68),443(3)	Complanaruside	-	+
**B27**	7.56	N	C_16_H_11_O_5_	283.06175	283.06158	3.229	MS^2^[283]:268(100),254(7),269(3),253(1),255(1),239(1),265(1)	Calycosin isomer	+	-
**B28**	7.69	P	C_16_H_13_O_5_	285.07630	285.07550	−0.877	MS^2^[285]:270(100),253(43),225(19),137(9),229(7),257(3),181(2) MS^3^[270]:137(100),253(58),214(29),213(15),134(11),242(11)	Calycosin isomer	+	-
**B29**	7.70	N	C_24_H_23_O_11_	487.12513	487.12631	3.793	MS^2^[487]:283(100),268(50),427(14),193(11),419(10),253(3)	Calycosin-7-O-β-D-glucoside-6″-*o*-acetate	-	+
**B30**	7.80	P	C_17_H_15_O_6_	315.08686	315.08594	−1.189	MS^2^[315]:300(100),283(19),255(7),167(5),301(5),259(4),138(3),269(3),168(2),297(1)	7,3′-dihydroxy-8,4-dimethoxyisoflavone/isomer	+	++
**B31**	7.87	N	C_16_H_11_O_4_	267.06683	267.06693	4.533	MS^2^[267]:252(100),253(5),249(2)	Formononetin isomer	+	++
		P	C_16_H_13_O_4_	269.08138	269.08215	4.886	MS^2^[269]:254(100),237(51),213(35),253(13),107(9),118(6),241(6),136(5)			
**B32**	7.89	P	C_26_H_27_O_11_	515.15534	515.15131	−3.751	MS^2^[515]:500(100),485(76),339(75),484(56),338(31),324(30),497(27),337(19),323(18)	Calycosin-7-O-β-D-glucoside-6″-o-butylene ester/isomer	+	-
**B33**	7.93	N	C_29_H_37_O_15_	625.21434	625.21527	4.116	MS^2^[625]:301(100),463(9),286(4),445(3),607(2),271(2),473(1)	Isomucronulatol-7,2′-di-o-glucoside/isomer	+	++
**B34**	8.12	N	C_16_H_11_O_5_	283.06175	283.06168	3.582	MS^2^[283]:268(100),269(3),255(1)	Calycosin isomer	-	+
**B35**	8.15	N	C_23_H_23_O_11_	475.12513	475.12598	3.245	MS^2^[475]:298(100),297(48),299(30),283(23),269(12),284(10),457(10),277(6)	7,3′-dihydroxy-8,4-dimethoxyisoflavone/isomer	+	-
**B36**	8.24	N	C_16_H_11_O_4_	267.06683	267.06702	4.870	MS^2^[267]:252(100),253(1) MS^3^[252]:223(100),208(65),224(54),132(21),195(15),196(5)	Formononetin isomer	+	++
**B37** *	8.26	P	C_22_H_23_O_9_	431.13421	431.13263	−2.386	MS^2^[431]:269(100),343(0.3),413(0.2)	Ononin	-	+
**B38**	8.41	N	C_17_H_15_O_5_	299.09305	299.09293	3.115	MS^2^[299]:284(100),269(1),255(1)	Pratensein/ isomer	++	+
		P	C_17_H_17_O_5_	301.10760	301.10641	−2.126	MS^2^[301]:167(100),269(26),191(21),147(19),163(12),273(11),207(9),286(6),241(6),270(3)			
**B39**	8.53	P	C_26_H_27_O_11_	515.15534	515.15076	−4.819	MS^2^[515]:339(100),321(3),199(1)	Calycosin-7-O-β-D-glucoside-6″-O-butylene ester	+	-
**B40**	8.54	N	C_16_H_15_O_5_	287.09305	287.09183	1.498	MS^2^[287]:272(100),135(93),165(46),177(29),121(22),147(19)	(3R)-7,2′,3′-trihydroxy-4′-methoxy isoflavonone/isomer	-	+
**B41**	8.70	N	C_29_H_37_O_15_	625.21434	625.21558	−4.782	MS^2^[625]:323(100),301(30),245(5),263(3),268(3),283(3),341(2),607(2)	Isomucronulatol-7,2′-di-o-glucoside/isomer	-	+
**B42**	8.73	N	C_23_H_23_O_11_	475.12513	475.12601	3.308	MS^2^[475]:299(100),284(62),298(18),297(17),283(9),285(9),269(9),151(1)	Odoratin-7-O-*β*-D-glucoside/isomer	+	-
**B43**	8.78	N	C_16_H_11_O_4_	267.06683	267.06696	4.158	MS^2^[267]:252(100),253(3),249(2),223(1)	Formononetin isomer	-	+
**B44**	8.96	N	C_17_H_15_O_5_	299.09305	299.09314	3.817	MS^2^[299]:284(100),269(4)	Pratensein/ isomer	+	++
	9.00	P	C_17_H_17_O_5_	301.10760	301.10641	−2.126	MS^2^[301]:167(100),269(22),191(20),147(15),163(10),273(9),207(7),241(6),286(2),270(2)			
**B45**	9.03	N	C_17_H_17_O_5_	301.10870	301.10770	3.479	MS^2^[301]:286(100),109(14),135(12),147(10),283(8),271(6),179(3),153(2),257(2)	(3R)-8,2′-Dihydroxy-7,4′-dimethoxy-isoflavan/isomer	++	+
	9.08	P	C_17_H_19_O_5_	303.12325	303.12225	−1.485	MS^2^[303]:167(100),149(32),123(19),284(16),181(14),168(7),219(6),270(5),193(5)			
**B46**	9.17	N	C_16_H_11_O_4_	267.06683	267.06689	3.046	MS^2^[267]:252(100),253(5) MS^3^[252]:223(100),208(70),224(46),132(16),195(15),196(7),179(3),225(2)	Formononetin isomer	+	-
		P	C_16_H_13_O_4_	269.08138	269.08182	3.659	MS^2^[269]:269(100),254(72),237(40),213(29),270(18),253(10),107(7),118(4),136(3) MS^3^[269]:254(100),253(32),214(11),163(7)			
**B47**	9.23	N	C_17_H_17_O_5_	301.10870	301.10880	3.811	MS^2^[301]:286(100),135(19),109(15),147(10),121(8),283(6),271(6),179(6) MS^3^[286]:271(100),242(8),268(5)	(3R)-8,2′-dihydroxy-7,4′-dimethoxy-isoflavan/isomer	-	+
	9.24	P	C_17_H_19_O_5_	303.12325	303.12219	−1.683	MS^2^[303]:167(100),149(29),123(22),181(16),193(6),285(2),219(1),168(1)			
**B48** *	9.25	N	C_23_H_27_O_10_	463.16152	463.16254	3.757	MS^2^[463]:301(100),299(1) MS^3^[301]:286(100)	Astraisoflavan-7-O-β-D- glucoside	+	++
**B49**		N	C_17_H_13_O_5_	297.07740	297.07748	3.823	MS^2^[297]:282(100),283(4),279(3),267(2),253(2),254(1),167(1)	Afromosin	++	+
	9.40	P	C_17_H_15_O_5_	299.09195	299.09119	−0.702	MS^2^[299]:284(100),166(23),243(21),239(11),267(11),285(10),137(4) MS^3^[284]:256(100),267(27),166(16),253(10),255(8),227(8),254(6),241(5)			
**B50**	9.42	N	C_23_H_27_O_10_	463.16152	463.16241	3.477	MS^2^[463]:287(100),272(3),395(3),213(1)	Astraisoflavan-7-O-β-D- glucoside isomer	+	-
**B51**	9.45	N	C_16_H_11_O_4_	267.06683	267.06699	4.757	MS^2^[267]:252(100),253(1) MS^3^[252]:223(100),208(73),224(49),132(21),195(13),196(4)	Formononetin isomer	-	+
		P	C_16_H_13_O_4_	269.08138	269.08035	−1.804	MS^2^[269]:269(100),254(63),237(33),213(25),270(20),253(11),107(7),118(5),136(5),241(3) MS^3^[269]:213(100),175(80),254(65),237(38),253(29),238(25)			
**B52** *	9.58	N	C_16_H_11_O_5_	283.06175	283.06183	4.112	MS^2^[283]:268(100),269(1) MS^3^[268]:240(100),211(64),239(62),224(45),195(30),184(26)	Calycosin	++	+
		P	C_16_H_13_O_5_	285.07630	285.07520	−1.929	MS^2^[285]:270(100),253(43),225(20),137(9),229(7),257(3),181(2),175(1) MS^3^[270]:137(100),253(50),214(35),134(13),213(13),242(10)			
**B53**	9.61	P	C_26_H_27_O_11_	515.15534	515.15076	−4.819	MS^2^[515]:339(100),500(7),199(7),353(1)	Calycosin-7-O-β-D-glucoside-6″-O-butylene ester/isomer	+	-
**B54**	9.83	N	C_17_H_17_O_5_	301.10870	301.10886	4.011	MS^2^[301]:286(100),109(17),135(12),147(8),271(7),283(7),259(3),121(3)	(3R)-8,2′-Dihydroxy-7,4′-dimethoxy-isoflavan/isomer	+	-
		P	C_17_H_19_O_5_	303.12325	303.12247	−0.759	MS^2^[303]:167(100),149(30),123(23),181(19),193(6) MS^3^[267]:152(100),134(29),139(11),167(9),124(8),167(2),106(2)			
**B55**	9.94	P	C_26_H_27_O_11_	515.15534	515.15472	−0.132	MS^2^[515]:500(100),339(59),215(21),501(18),324(10),357(8),340(7) MS^3^[500]:324(100),342(75),425(68),383(44),485(17),	Calycosin-7-O-β-D-glucoside-6″-O-butylene ester/isomer	+	-
**B56**	10.00	P	C_17_H_15_O_6_	315.08686	315.08588	−1.379	MS^2^[315]:300(100),271(49),283(21),287(13),138(12),259(11),199(9),255(7)	7,3′-dihydroxy-8,4-dimethoxyisoflavone/isomer	++	+
**B57**	10.04	N	C_17_H_15_O_5_	299.09305	299.09329	4.319	MS^2^[299]:284(100),269(4) MS^3^[284]:269(100)	Pratensein/ isomer	-	+
**B58**	10.25	N	C_17_H_15_O_5_	299.09305	299.09323	4.118	MS^2^[299]:284(100),269(6),255(6),165(4),271(4)	Pratensein/ isomer	+	++
**B59**	10.29	N	C_23_H_23_O_11_	475.12513	475.12598	5.245	MS^2^[475]:299(100),341(5),323(4),165(3),429(2),397(2),271(2)	Odoratin-7-O--D-glucoside	+	-
		P	C_23_H_25_O_11_	477.13969	477.13809	−2.196	MS^2^[477]:301(100),401(46),199(26),269(18),458(15),405(14)			
**B60**	10.40	N	C_17_H_17_O_5_	301.10870	301.10886	4.011	MS^2^[301]:286(100),135(38),121(17),109(13),147(10),283(8),179(7),271(6) MS^3^[286]:271(100),242(10),268(7),269(6)	(3R)-8,2′-dihydroxy-7,4′-dimethoxy-isoflavan/ isomer	-	+
		P	C_17_H_19_O_5_	303.12325	303.12247	−0.759	MS^2^[303]:167(100),149(29),123(23),181(16),193(7),285(2),261(1),167(1)			
**B61**	10.99	N	C_16_H_11_O_4_	267.06683	267.06693	4.533	MS^2^[267]:252(100),253(5),249(2)	Formononetin isomer	-	+
		P	C_16_H_13_O_4_	269.08138	269.08191	3.994	MS^2^[269]:254(100),237(42),213(35),66(13),253(12),107(10),118(5)			
**B62**	11.75	P	C_17_H_17_O_5_	301.10760	301.10690	−0.499	MS^2^[301]:167(100),269(22),191(20),147(16),163(10),273(10),207(7),241(6),207(4)	Pratensein/ isomer	-	+
**B63** *	12.20	N	C_16_H_11_O_7_	315.04992	315.05081	2.796	MS^2^[315]:300(100),301(3)	Isorhamnetin	++	+
**B64** *	13.99	N	C_16_H_11_O_4_	267.06683	267.06699	4.757	MS^2^[267]:252(100),253(3)	Formononetin	++	+
		P	C_16_H_13_O_4_	269.08138	269.08041	−1.581	MS^2^[269]:254(100),251(65),237(47),213(32),253(12),107(8)			
**B65**	14.79	P	C_17_H_17_O_5_	301.10760	301.10666	−1.296	MS^2^[301]:269(100),167(98),147(61),191(56),273(36),163(32),241(26),207(22),270(10)	Pratensein/ isomer	++	+

Note: *: Compared with the reference standards; +: detected; -: undetected; ++: more abundant.

**Table 3 molecules-24-02948-t003:** Chemical information of identified saponins in AR and FAR.

No	Name	Formula	Core structure	Substituent group
**1**	Astragaloside I	C_45_H_72_O_16_	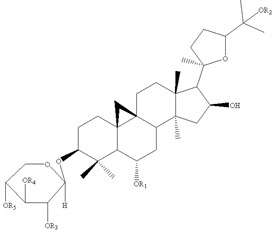	R_1_=glu R_2_=R_5_=H R_3_=R_4_=Ac
**2**	Isoastragaloside I	C_45_H_72_O_16_		R_1_=glu R_2_=R_4_=H R_3_=R_5_=Ac
**3**	Astragaloside II	C_43_H_70_O_15_		R_1_=glu R_2_=R_4_=R_5_=H R_3_=Ac
**4**	Isoastragaloside II	C_43_H_70_O_15_		R_1_=glu R_2_=R_3_=R_5_=H R_4_=Ac
**5**	Astragaloside III	C_41_H_68_O_14_		R_1_=R_2_=glu R_5_=H R_3_=R_4_=Ac
**6**	Astragaloside IV	C_41_H_68_O_14_		R_1_=glu R_2_=R_3_=R_4_=R_5_=H
**7**	Isoastragaloside IV	C_41_H_68_O_14_		R_1_=R_3_=R_4_=R_5_=H R_2_=glu
**8**	Acetylastragaloside I	C_47_H_74_O_17_		R_1_=glu R_2_=H R_3_=R_4_=R_5_=Ac
**9**	Agroastragaloside III	C_51_H_82_O_21_		R_1_=R_2_=glu R_5_=H R_3_=R_4_=Ac
**10**	Malonylastragaloside I	C_48_H_74_O_19_		R_1_=glu R_2_=H R_3_=R_4_=Ac R_5_=malonyl
**11**	Astramembranoside A	C_42_H_70_O_15_	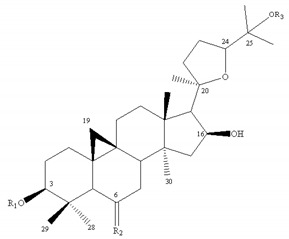	R_1_=H R_2_=α-O-glu β-H R_3_=glu
**12**	MongHolicoside A	C_36_H_62_O_11_	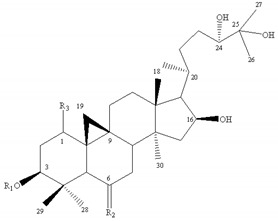	R_1_=glu R_2_=α-OH β-H R_3_=OH
**13**	MongHolicoside B	C_36_H_60_O_11_		R_1_=glu R_2_=O R_3_=OH
**14**	Agroastragaloside I	C_45_H_74_O_16_	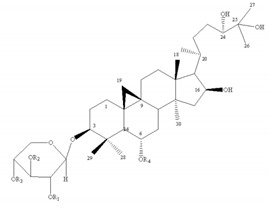	R_1_=R_2_=Ac R_3_=H R_4_=glu
**15**	Agroastragaloside II	C_43_H_72_O_15_		R_1_=Ac R_2_=R_3_=H R_4_=glu
**16**	CyclocantHoside E	C_41_H_70_O_14_		R_1_=R_2_=R_3_=H R_4_=glu
**17**	Mongholicoside II	C_38_H_62_O_11_	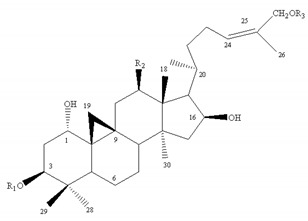	R_1_= COCH_3_ R_2_=OH R_3_=glu
**18**	Huangqiyenin E	C_42_H_66_O_14_	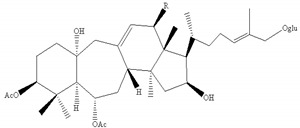	R=OAc
**19**	Soyasaponin I	C_48_H_78_O_18_	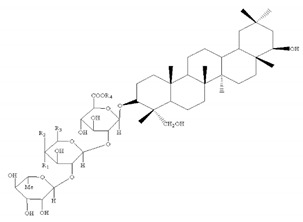	R_1_=R_4_=H R_2_=OH R_3_=CH_2_OH

**Table 4 molecules-24-02948-t004:** Chemical information of identified flavonoids in AR and FAR.

No	Name	Formula	Core structure	Substituent group
**1**	Isorhamnetin	C_16_H_12_O_7_	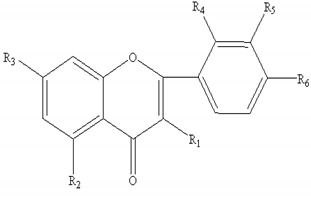	R_1_=OH R_2_=R_3_=OH R_4_=H R_5_=OCH_3_ R_6_=OH
**2**	Kaempferol-4′-methylether-3-β-D-glucoside	C_22_H_22_O_11_		R_1_=O-glu R_2_=R_3_=OH R_4_=R_5_=H R_6_=OCH_3_
**3**	Isorhamnetin-3-β-D- glucoside	C_22_H_22_O_12_		R_1_=O-glu R_2_=R_3_=R_6_=OH R_4_=H R_5_=OCH_3_
**4**	Neocomplanoside	C_24_H_24_O_12_		R_1_=O-(6-O-acetyl)-glu R_4_=R_5_=H R_3_=OCH_3_R_2_=R_6_=OH
**5**	Complanaruside	C_28_H_32_O_16_		R_1_=R_6_=O-glu R_2_=OH R_3_=OCH_3_ R_4_=R_5_=H
**6**	(3R)-7,2′,3′-trihydroxy- 4′-methoxy isoflavonone	C_16_H_16_O_5_	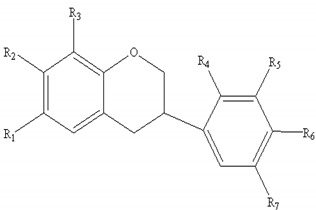	R_1_=R_3_=R_7_=H R_2_=R_4_=R_5_=OH R_6_=OCH_3_
**7**	(3R)-8,2′-dihydroxy-7,4′-dimethoxy-isoflavan	C_17_H_18_O_5_		R_1_=R_5_=R_7_=H R_2_=R_6_=OCH_3_ R_3_=R_4_=OH
**8**	Astraisoflavan-7-O-β-D- glucoside isomer	C_23_H_28_O_10_		R_1_=R_3_=R_7_=H R_2_=O-glu R_4_=OH R_5_=R_6_=OCH_3_
**9**	5′-hydroxy isomucronulatol 2′,5′-di-O-glucoside	C_29_H_38_O_16_		R_1_=R_3_=H R_2_=OH R_4_=R_7_=O-glu R_5_=R_6_=OCH_3_
**10**	Isomucronulatol-7,2′-di-O- glucoside	C_29_H_38_O_15_		R_1_=R_3_=R_7_=H R_2_=R_4_=O-glu R_5_=R_6_=OCH_3_
**11**	Formononetin	C_10_H_12_O_4_	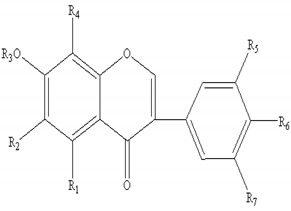	R_1_=R_2_=R_3_=R_4_=R_5_=R_7_=H R_6_=OCH_3_
**12**	5,7,4′-trihydroxy- isoflavonone	C_15_H_10_O_5_		R_1_=R_6_=OH R_2_=R_3_=R_4_=R_5_=R_7_=H
**13**	Calycosin	C_16_H_12_O_5_		R_1_=R_2_=R_3_=R_4_=R_7_=H R_5_=OH R_6_=OCH_3_
**14**	Afromosin	C_17_H_14_O_5_		R_1_=R_3_=R_4_=R_5_=R_7_=H R_2_=R_6_=OCH_3_
**15**	7,3′-dihydroxy-8,4′- dimethoxyisoflavone	C_17_H_14_O_6_		R_1_=R_2_=R_3_=R_7_=H R_4_=R_6_=OCH_3_ R_5_=OH
**16**	Ononin	C_22_H_22_O_9_		R_1_=R_2_=R_4_=R_5_=R_7_=H R_3_=glu R_6_=OCH_3_
**17**	Genistin	C_21_H_20_O_10_		R_1_=R_6_=OH R_2_=R_4_=R_5_=R_7_=H R_3_=glu
**18**	Calycosin-7-O-β-D- glucoside	C_22_H_22_O_10_		R_1_=R_2_=R_4_=R_7_=H R_3_=glu R_5_=OH R_6_=OCH_3_
**19**	Calycosin-7-O-β-D- glucoside-6″-O-acetate	C_24_H_24_O_11_		R_1_=R_2_=R_4_=R_7_=H R_3_=6″-acetate-O-glu R_5_=OH R_6_=OCH_3_
**20**	Calycosin-7-O-β-D-glucoside-6″-O-butylene ester	C_26_H_26_O_11_		R_1_=R_2_=R_4_=R_7_=H R_3_=6″-butylene ester-O-glu R_5_=OH R_6_=OCH_3_
**21**	Odoratin-7-O-β-D- glucoside	C_23_H_24_O_11_		R_1_=R_4_=R_7_=H R_2_=R_6_=OCH_3_ R_3_=glu R_5_=OH

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
