# Peer review of "Chemical Constituent Profiling of Paecilomyces cicadae Liquid Fermentation for Astragli Radix"

_molecules, 2019, doi:10.3390/molecules24162948_

Round 1

Reviewer 1 Report

Manuscript (molecules-575472) entitled „Chemical constituents profiling of Paecilomyces cicadae liquid fermentation for Astragli Radix“ was aimed to obtain a comprehensive knowledge chemical constituents from the fermented Astragli Radix by ultra-high performance liquid chromatography coupled with high-resolution mass spectrometry (UHPLC-HRMS). From experimentation to data evaluation, everything is well organized and clearly described and the LC-MS analysis appears to be carefully performed. In my opinion, the quality of this manuscript is acceptable to be published in Molecules, after minor revision. Some remarks are summarized as follows:

Table 1 and 2  MS3 data is missing in these tables. Only a few columns in the tables have data for the MS3 spectrum of identified compounds. I think it would be very useful to display as much MS data as possible because they are very conducive to structural elucidation.

Table 2           Compound B63 is marked as isoquercitrin and confirmed by standard. Mass of isoquercitrin (quercetin 3-O-glucoside) is 463 m/z in negative ion mode. This is a mistake; it is another compound, perhaps methylated quercetin. Please correct it and check the other compounds in Table 1 and 2. In this case, the MS3 spectrum would provide further fragmentation data.

Author Response

Response to Reviewer 1 Comments

Thanks for the reviewers’ and editor’ comments concerning our manuscript entitled Chemical constituents profiling of Paecilomyces cicadae liquid fermentation for Astragli Radix (Manuscript ID: molecules-575472 ). Those comments are valuable and helpful for revising and improving our manuscript. We have studied the comments carefully and revised our manuscript. The main corrections in the paper and the responds to the comments were listed as follows:

-Reviewer

Manuscript (molecules-575472) entitled “Chemical constituents profiling of Paecilomyces cicadae liquid fermentation for Astragli Radix” was aimed to obtain a comprehensive knowledge chemical constituents from the fermented Astragli Radix by ultra-high performance liquid chromatography coupled with high-resolution mass spectrometry (UHPLC-HRMS). From experimentation to data evaluation, everything is well organized and clearly described and the LC-MS analysis appears to be carefully performed. In my opinion, the quality of this manuscript is acceptable to be published in Molecules, after minor revision. Some remarks are summarized as follows:

Point 1: Table 1 and 2 MS3 data is missing in these tables. Only a few columns in the tables have data for the MS3 spectrum of identified compounds. I think it would be very useful to display as much MS data as possible because they are very conducive to structural elucidation.

Response: Thanks for the suggestions you kindly offered. We have added some MS3 spectra of the identified compounds. For the limit of the instrument, the ESI-MS3 spectra of some target constituents were lacked. However, according to the obtained accurate molecular weight, combined with chromatographic retention behaviors, mass spectrometry cleavage, characteristic fragment ions, references comparison and related literature reports, most of those compounds could be positively or tentatively characterized.

Point 2: Compound B63 is marked as isoquercitrin and confirmed by standard. Mass of isoquercitrin (quercetin 3-O-glucoside) is 463 m/z in negative ion mode. This is a mistake; it is another compound, perhaps methylated quercetin. Please correct it and check the other compounds in Table 1 and 2. In this case, the MS3 spectrum would provide further fragmentation data

Response: Thanks for the suggestions you kindly offered. Base on the reference comparison, we have revised compound B63 as isorhamnetin, accurately.

Reviewer 2 Report

Wang et al reported the change of metabolites in Astragli Radix (AR) fermented by Paecilomyces cicadae.  UHPLC-LTQ-Orbitrap MS was used to profile AR and FAR (fermented AR). A total 107 metabolites were identified from both AR and FAR. These compounds were either flavonoids (65) or saponins (42). Significant differences were observed between FAR and AR. Some metabolites were only observed in FAR and some were exclusively found in AR. The metabolites were identified by matching to reference compounds and/or by tandem MS in both positive and negative modes. Overall the manuscript is well written and straightforward. However, I have some questions/comments as below.

Core structures of representative flavonoid and astragalus saponin (cycloastragenol and oleanane), and their numbering should be given to help readers understand the metabolite structures. Loss of 162 (hexose), 132 (pentose) and other neutral losses (such as CO, CO2 and H2O etc.) were used to assist in spectral analysis and structure elucidation. However, loss of 162 and 132 are characteristic of O-linked hexose and pentose. C-linked flavonoid glycosides are prevalent in plants and the loss of their carbohydrate moieties is different from that of the O-linked glycosides (Lei et Anal. Chem. 2015, 87, 7373−7381). Why did the authors not consider losses of these C-linked carbohydrates in their DPI and NLF approaches?  The identification would not be complete if C-linked glycosides were not considered. The author reported qualitative difference between AR and FAR. But no quantitative information is lacking. Even semi quantitative information is helpful. For example, if one compound is more abundant in AR than in AFR, the authors can use +++ for AR and + for FAR in the table. For the metabolites that were identified by MSMS without reference standards, were they identified by searching MS/MS spectral libraries? If yes, matching scores should be given. If not, supplemental figures should be provided to illustrate the fragmentation patterns of the metabolites that explain the observed fragments. Most of the fragments observed for astragalus saponin are derived from loss of carbohydrates, acetyl and water. While these losses are instrumental in determining the presence or absence of these functional groups/substituents, their positions are difficult to assign unless the core structures (ie, cycloastragenol and oleanane) undergoes fragmentation. But their fragmentation was not reported in the manuscript. Did the authors observe the core structure fragmentation?

Author Response

Response to Reviewer 1 Comments

Thanks for the your kindly comments concerning our manuscript entitled Chemical constituents profiling of Paecilomyces cicadae liquid fermentation for Astragli Radix (Manuscript ID: molecules-575472 ). Those comments are valuable and helpful for revising and improving our manuscript. We have studied the comments carefully and revised our manuscript. The main corrections in the paper and the responds to the comments were listed in the follow file.
